# Coupling Coordination and Influencing Factors of Land Development Intensity and Urban Resilience of the Yangtze River Delta Urban Agglomeration

**Xiang Luo** [1], **Chao Cheng** [1], **Yue Pan** [2,*] and **Tiantian Yang** [1]

[1] School of Public Administration, Central China Normal University, Wuhan 430070, China; philiplaw@163.com (X.L.); frankc0602@163.com (C.C.); ytt6927@163.com (T.Y.)
[2] School of Civil Engineering and Architecture, Wuhan Institute of Technology, Wuhan 430070, China
[*] Correspondence: py95351721@163.com; Tel.: +86-137-0717-8556

**Abstract:** The rapid urbanization of the Yangtze River Delta urban agglomeration has led to the convergence of population, land and capital. The coordination between land development intensity and urban resilience has become a key issue in the post-urbanization period. From the perspective of regional overall and coordinated development, we constructed an evaluation index system of land development intensity and urban resilience. Then, the comprehensive evaluation model, coupling coordination degree model and panel Tobit regression model were adopted to systematically study the temporal and spatial differentiation of and influencing factors on the coupling coordination degree between land development intensity and urban resilience in the Yangtze River Delta urban agglomeration from 2009 to 2019. The results show that from 2009 to 2019, the land development intensity exhibited a slow and fluctuating increase, while the urban resilience displayed continuous growth, and the level of land development intensity generally lagged behind that of urban resilience. From 2009 to 2019, the average coupling coordination degree between land development intensity and urban resilience in the Yangtze River Delta urban agglomeration increased from 0.5177 to 0.6626, which generally changed from bare coordination to moderate coordination. In terms of spatial distribution, the coastal cities and cities along the Yangtze River were characterized by high coupling coordination degrees, which formed a "T" shape distribution pattern. In addition, the coupling coordination types showed certain spatio-temporal heterogeneity among cities. Finally, land economic benefit, green industrial development, scientific and technological innovation, social management and infrastructure all had significant impacts on the coupling and coordination between land development intensity and urban resilience in the Yangtze River Delta urban agglomeration.

**Keywords:** land development intensity; urban resilience; coupling coordination; panel tobit regression model

## 1. Introduction

A city is a highly complex giant system, whose safe operation is critical for the sustainable development of the whole of society. Since the reform and opening-up, China's urbanization has continuously accelerated, and the urbanization rate reached about 60.60% in 2019. On the one hand, rapid urbanization promotes the convergence of population, capital and other factors in cities and the rapid expansion of construction land. On the other hand, the high convergence of various production factors and the scarcity of land resources promote continuous increases in land development intensity. Undoubtedly, the urban system is becoming more complex, and the uncertainty risks faced by cities are also increasing. In recent years, extreme natural disasters such as extreme cold, rainstorms and high temperatures have frequently occurred all over the world. Cases of urban paralysis caused by various black-swan events such as chemical

leakage, municipal system failure and public health crises occur frequently. For instance, the outbreak of COVID-19 in 2019 led to the functional failure of many cities for a time around the world, causing enormous losses in economic and social development and human life safety. City vulnerability is a crucial factor that restricts the sustainable development of a city in the process of urbanization. The construction of urban resilience systems has become an important part of the national governance system in China [1]. In the Guidelines on Establishing and Supervising the Implementation of Territorial Space Planning System issued in May 2019 and the Proposal on formulating the 14th Five-Year Plan for National Economic and Social Development and the Long-term Goals for 2035 promulgated in October 2020 and other documents, great emphasis is laid on the construction of resilient cities and the improvement of disaster prevention awareness. Urban agglomerations are important engines of China's new urbanization, and they are also the regions with the most significant differentiation of urban land development intensity. A sustainable and healthy development model is crucial for the coordinated development of urban agglomerations. Therefore, in the post-urbanization period, it is the strategic requirement of current new urbanizations and an urgent need in the construction of sustainable and safe cities to integrate resilience into land development and utilization in urban agglomeration and promote coordination between land development intensity and urban resilience [2].

The term "resilience" originated from mechanical engineering, which was introduced into ecosystem research by the Canadian ecologist Holling in 1973 [3] and then gradually applied to studies of urban safety and disaster prevention and reduction. The concept of "resilience" has undergone three stages of development from "engineering resilience" to "ecological resilience" and then to "evolutionary resilience" [4], and the latter two stages have laid the theoretical basis for current research on resilient cities. The phrase resilient city refers to a city that can maintain or quickly recover the required functions, adapt to changes when faced with external shocks and disturbance and rapidly transform the system that limits the current or future adaptability [5]. The theory of resilient cities provides a new perspective and path for coping with the threat of urban uncertainties and promoting the sustainable development of humans and land. Currently, the research on urban resilience is mainly based on the "adaptive cycle" theory [6] proposed by Gunderson and Holling. From different perspectives, extensive research has been carried out on the framework of resilient cities [7,8], the capacity and characteristics of resilient cities [9,10], the quantitative evaluation methods of urban resilience index [11,12] and the promotion strategy of urban resilience [13]. From the aspect of the framework of resilient cities, the Resilience Alliance proposed four priority areas for urban resilience research, including urban metabolic flow, social dynamics, governance network and construction environment [14]; The Rockefeller Foundation proposed the research framework of urban resilience, covering four dimensions, namely health and well-being, economy and society, infrastructure and environment and leadership and strategy [8]; scholars such as Jha (2013), Zhang (2019) and Chen (2020) divided the urban resilience system into subsystems including ecological resilience, economic resilience, social resilience and engineering resilience [15–17]. In fact, a large number of existing studies emphasize different aspects of urban resilience from the dimensions of ecology, economy, society and engineering, laying the foundation for the framework of urban resilience research in this paper. In addition, different methods have been proposed for the quantitative evaluation of urban resilience, including the Comprehensive Evaluation Model [18], the Delphi Method and Cloud Model [19], the System Dynamics Model [20] and some other evaluation methods. With the continuous enrichment of theories and methods, numerous studies have been carried out regarding urban resilience assessment and response strategies under different scenarios such as economic crisis [21], rain and flood disaster [22], climate change [23] and public security events [24].

The urban resilience system is closely related to other urban systems. Some previous studies have explored the coupling coordination relationship between urban resilience

and urbanization [25]. Some scholars have also studied the coupling coordination and interaction mechanism from various perspectives such as urban resilience with a smart city system [26], urban land evolution [27], land use efficiency [28], economic level [29] and industrial development [30]. Previous studies have revealed the interaction of different urban systems with the urban resilience system. However, it should be noted that land is the direct carrier of urban systems, and changes in land development and utilization are the internal driving force for the evolution of the relationship between urban systems [31]. As a measure of land development and utilization, land development intensity has a close mutual restriction and promotion relationship with the urban resilience system. The development of land development intensity and urban resilience is a long-term dynamic process. On the one hand, higher urban resilience can enhance the ability of a city to resist risks and support high-intensity land development and utilization; on the other hand, with increasing land development intensity, population convergence and high-intensity construction will increase the pressure of a city to resist risks and reduce its resilience. However, the economic benefits and capital investment brought by high-intensity land development (namely the increase in land economic density) also lay a solid foundation for the improvement of urban resilience (Figure 1). In view of this, based on the internal relationship between land development intensity and urban resilience, firstly, we constructed the comprehensive evaluation index system of these two components by referring to the existing relevant literature. Secondly, we analyzed spatio-temporal differentiation characteristics of land development intensity and urban resilience in the Yangtze River Delta urban agglomeration from 2009 to 2019 by using the comprehensive evaluation model, coupling coordination degree model and panel Tobit regression model and also explored the coupling coordination relationship between them and the influencing factors. Eventually, the relevant policy suggestions were put forward according to research results. There are two research purposes of this paper. On the one hand, it provides supporting methods for the scientific measurement of the coupling and coordination relationship between land development intensity and urban resilience. On the other hand, it provides a decision-making basis for the promotion of efficient and intensive land use as well as the safe and sustainable development of cities in the Yangtze River Delta urban agglomeration.

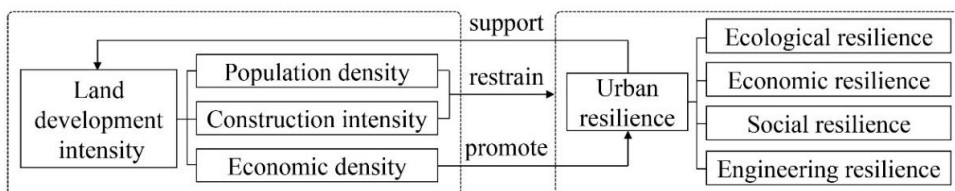

**Figure 1.** Coupling coordination relationship between land development intensity and urban resilience.

## 2. Research Area, Data Sources and Research Methods

### 2.1. Overview of the Research Area

In accordance with the development plan of the Yangtze River Delta urban agglomeration, the research area of this paper covers three provinces and one city in the Yangtze River Delta, including 26 cities, namely Shanghai, Nanjing, Wuxi, Changzhou, Suzhou, Nantong, Yancheng, Yangzhou, Zhenjiang, Taizhou, Hangzhou, Ningbo, Jiaxing, Huzhou, Shaoxing, Jinhua, Zhoushan, Taizhou, Hefei, Wuhu, Ma'anshan, Tongling, Anqing, Chuzhou, Chizhou and Xuancheng (Figure 2). The land area of the Yangtze River Delta urban agglomeration is 211,700 km$^2$, accounting for about 2.2% of the total land area, nearly 20% of the economic aggregate and more than 10% of the permanent resident population of China. It is one of the most important functional areas supporting and leading China's economic development. By 2019, the population urbanization rate of the

Yangtze River Delta urban agglomeration reached 68.4%; the urban construction land area in the municipal district reached about 8600 km²; the regional GDP reached 19.7 trillion yuan. Population, land and capital are highly converged in this area, promoting continuous increases in land development intensity. Under the transition from the pursuit of high speed to the pursuit of high quality in urbanization, it is of great significance to explore the spatial differentiation and influencing factors of the coordinated relationship between land development intensity and urban resilience in this area. In addition, the temporal and spatial variations of land development intensity of cities in the Yangtze River Delta are significant. The clarification of the coordination relationship between land development intensity and urban resilience can help to identify weaknesses in urban risk prevention and control, improve the ability of cities to cope with sudden disasters and provide an important reference for the construction of a safe, resilient and sustainable urban agglomeration.

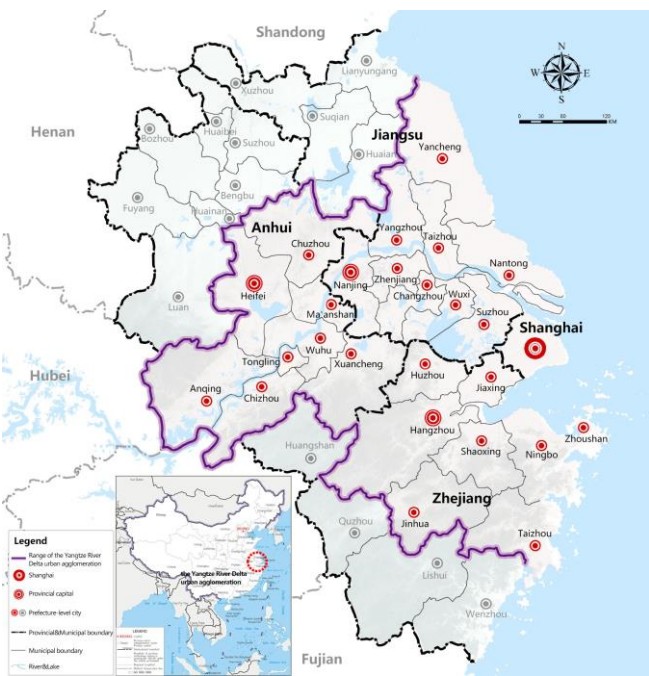

**Figure 2.** Location map of the Yangtze River Delta urban agglomeration.

*2.2. Data Sources*

This research includes the data of land development intensity and urban resilience of 26 cities in the Yangtze River Delta from 2009 to 2019. All data and indicators were derived from the China Statistical Yearbook of Cities, the China Statistical Yearbook of Urban Construction and the statistical yearbooks and statistical bulletins of provinces and cities in the Yangtze River Delta region. Individual outliers and missing data were represented by the average value of adjacent years or the province where they are located. The data of boundaries between administrative divisions at all levels were obtained from the 1:4 million National Basic Geographic Information Database (https://www.resdc.cn, accessed on 6 November 2021).

*2.3. Research Methods*

2.3.1. Construction of the Evaluation Index System

Urban land development intensity refers to the level of comprehensive utilization of urban land, which is generally measured by the ratio of urban construction land area to total urban area. However, this single-index measurement method cannot fully reflect the degree of land development and the bearing capacity of land for population, economy and society. Based on the existing literature related to the measurement of land

development intensity [32,33], we constructed an evaluation system to measure the land development intensity from three aspects: population intensity, construction intensity and economic benefits. The specific indicators are shown in Table 1.

**Table 1.** Evaluation index system of land development intensity and urban resilience.

| Target Layer | System Layer | Evaluation Indicator | Index Meaning | Weight | Attribute |
|---|---|---|---|---|---|
| **Evaluation index system of land development intensity level** | Population density | Population density in municipal district (person/km²) | Population carrying capacity | 0.2811 | + |
| | Construction intensity | Proportion of construction land area in municipal district (%) | Land construction intensity | 0.3647 | + |
| | Economic density | GDP per square kilometer land in municipal district (10,000 yuan/km²) | Land economic benefits | 0.3542 | + |
| **Evaluation index system of urban resilience level** | Ecological Resilience (0.0925) | Green space rate of built-up area (%) | Urban greening | 0.0180 | + |
| | | Green space area of parks per capita (m²) | Disaster avoidance space | 0.0350 | + |
| | | Industrial wastewater emission per 10,000 yuan GDP (t) | Industrial green development level | 0.0145 | - |
| | | Industrial SO₂ emission per 10,000 yuan GDP (kg) | | 0.0064 | - |
| | | Urban sewage treatment rate (%) | Environmental governance response | 0.0038 | + |
| | | Comprehensive utilization rate of industrial solid waste (%) | Comprehensive utilization of waste | 0.0109 | + |
| | | Harmless treatment rate of domestic waste (%) | Environmental remediation | 0.0039 | + |
| | Economic Resilience (0.2879) | Per capita GDP (yuan) | Economic development | 0.0608 | + |
| | | Proportion of secondary and tertiary industry GDP (%) | Industrial structure l | 0.0126 | + |
| | | Proportion of government public financial revenue in GDP (%) | Economic growth | 0.0396 | + |
| | | Deposit balance of per capita financial institutions (yuan) | Economic recovery capability | 0.0934 | + |
| | | Proportion of science and technology expenditure in financial expenditure (%) | Scientific and technological innovation | 0.0815 | + |
| | Social Resilience (0.2838) | Average wage of employees (yuan) | Residents' ability to resist risk | 0.0563 | + |
| | | Registered unemployment rate in cities and towns (%) | Social employment pressure | 0.0402 | - |
| | | Number of doctors per 10,000 people (person) | Medical and health | 0.0421 | + |
| | | Number of beds in hospitals and health centers per 10,000 people (number) | | 0.0516 | + |
| | | Number of buses per 10,000 people (vehicles) | Transportation facilities | 0.0602 | + |
| | | Number of public management and social organization personnel per 10,000 people (person) | Social management | 0.0334 | + |
| | Engineering resilience (0.3486) | Per capita daily domestic water consumption (liters) | Efficiency of resource utilization | 0.0293 | - |
| | | Urban road area per capita (m²) | Road traffic | 0.0345 | + |
| | | Road network density in built-up area(km/km²) | | 0.0892 | + |
| | | Density of water supply pipeline in built-up area (km/km²) | Infrastructure | 0.1018 | + |
| | | Density of drainage pipeline in built-up area (km/km²) | | 0.0343 | + |
| | | Number of mobile phone users per 10,000 people (households) | Popularity of communication technology | 0.0595 | + |

The evaluation of urban resilience in this paper was performed mainly based on the research of Jha (2013), Chen (2020) and some other scholars [15,17]. The urban resilience system is divided into four subsystems, including ecological resilience, economic resilience, social resilience and engineering resilience. By referring to the existing literature [34,35] and indicators of sustainable urban development, an evaluation index system of urban resilience was established. Among the four subsystems, ecological resilience is the basis of urban sustainable development, reflecting the service function of an urban ecosystem and the level of green development. We selected seven indicators as the measurement standards, including the green space rate of built-up area, green space area of parks per capita, industrial wastewater emission per 10,000 yuan GDP, industrial SO₂ emission per 10,000 yuan GDP, urban sewage treatment rate, comprehensive utilization rate of industrial solid waste and harmless treatment rate of domestic waste. Economic resilience can represent the ability of a city to cope with market shocks as well

as its economic bearing capacity and recovery capacity after sudden disasters. This paper selected five indicators as the measurement standards, including per capita GDP, the proportion of secondary and tertiary industry GDP, the proportion of government public financial revenue in GDP, the deposit balance of per capita financial institutions, and the proportion of science and technology expenditure in government financial expenditure. Social resilience indicates the adaptive ability of the government, society, individuals and public welfare to risk disasters. This paper selected six indicators for the measurement, including the average wage of employees, the registered unemployment rate in cities and towns, the number of doctors per 10,000 people, the number of beds in hospitals and health centers per 10,000 people, the number of buses per 10,000 people and the number of public management and social organization personnel per 10,000 people. Engineering resilience reflects the ability of a city to resist, absorb and recover from risks through the level of urban infrastructure construction. This paper selected six indicators for the measurement, including daily domestic water consumption per capita, urban road area per capita, road network density, water supply pipeline density and drainage pipeline density in the built-up area and the number of mobile phone users per 10,000 people. The specific indicators are shown in Table 1.

### 2.3.2. Comprehensive Evaluation Model

In this paper, a comprehensive evaluation model was used to calculate the land development intensity and urban resilience level of the Yangtze River Delta urban agglomeration. The specific steps are as follows.

First, the indices were standardized. Considering the different dimensions and magnitudes of the selected indices, the max–min method was employed to standardize the values of each index.

Second, the weight of each index was calculated. The entropy weight method was adopted to determine the weight of each index. This method calculates the contribution rate of each index based on its information amount, effectively avoiding the subjectivity of weight assignment. The weight of each index is presented in Table 1.

Finally, the comprehensive index of land development intensity and urban resilience was calculated with the following formula:

$$S_1 = \sum_{i=1}^{n} w_i \times x_{ij} \qquad S_2 = \sum_{i=1}^{n} w_i' \times x_{ij}' \tag{1}$$

In this formula, $S_1$ represents the land development intensity index, $S_2$ represents the urban resilience index, $w_i$ and $w_i'$ represent the weight of index $i$, $x_{ij}$ and $x_{ij}'$ represent the standardized value of the $i$-th index in the $j$-th year of the land development intensity and urban resilience system.

### 2.3.3. Coupling Coordination Degree Model

In physics, coupling refers to the process of interaction and influence among multiple systems. This study used the coupling degree model in physics as a reference to measure the interaction strength between land development intensity ($S_1$) and urban resilience ($S_2$) with the following formula.

$$C = \left[ S_1 \times S_2 \bigg/ \left( S_1 + S_2 \big/ 2 \right)^2 \right]^{1/2} \tag{2}$$

In the formula, $S_1$ is the land development intensity index; $S_2$ is the urban resilience index; $C$ is the coupling degree, with values ranging from 0 to 1.

The coupling degree model can only indicate the presence or absence of interaction among systems but cannot quantify the level of coupling coordination. Therefore, a coupling coordination degree model was established to measure the status and level of coordinated development between the two systems. The calculation formula is as follows:

$$D = \sqrt{C \times T} \qquad T = \alpha S_1 + \beta S_2 \tag{3}$$

In this formula, $T$ is the comprehensive development evaluation index of land development intensity and urban resilience; $\alpha$ and $\beta$ are undetermined coefficients, with $\alpha + \beta = 1$. This paper assumed that in the interaction, land development intensity and urban resilience are equally important; therefore, the value was selected as $\alpha = \beta = 0.5$. $D$ refers to the coupling coordination degree with the values ranging from 0 to 1. Based on the existing research [36], the coupling coordination degree could be divided into five levels as follows: serious disorder (0.0–0.2), moderate disorder (0.2–0.4), bare coordination (0.4–0.6), moderate coordination (0.6–0.8) and excellent coordination (0.8–1.0). In addition, according to the relative relationship between land development intensity system ($S_1$) and urban resilience system ($S_2$), when $S_1 > S_2$, the coupling coordination is characterized by the lagging behind of urban resilience and vice versa.

### 2.3.4. Panel Tobit Regression Model

The value of the coupling coordination index was calculated to range from 0 to 1, which is a limited dependent variable. Traditional OLS regression tends to produce erroneous results, which can be overcome by the panel Tobit regression model. Therefore, the panel Tobit regression model was adopted to analyze the influencing factors on the coupling degree between land development intensity and urban resilience in the Yangtze River Delta urban agglomeration. The specific model is as follows:

$$y_{it} = \begin{cases} y_{it}^* = a_0 + \sum_{t=1}^{n} \beta_k x_{it} + \varepsilon_{it} & y_{it}^* > 0 \\ 0 & y_{it}^* \leq 0 \end{cases} \tag{4}$$

In the formula, $y_{it}$ represents the explained variable, $x_{it}$ indicates the explanatory variable, $\beta_k$ is the regression coefficient of the explanatory variable, $a_0$ is the constant term and $\varepsilon_{it}$ is the random error term following the $N(0, \sigma^2)$ distribution.

## 3. Results Analysis

### 3.1. Overview of Land Development Intensity and Urban Resilience

According to the evaluation index system in Table 1, the entropy weight method was first used to calculate the index weights. Then, the land development intensity and urban resilience of the Yangtze River Delta urban agglomeration from 2009 to 2019 were calculated with the comprehensive evaluation model (Figure 3). To better analyze the spatial pattern and dynamic evolution of the coupling coordinated development of land development intensity and urban resilience, we used the ArcGIS10.6 software and visualized the levels of land development intensity and urban resilience in 2009, 2014 and 2019 (Figures 4 and 5).

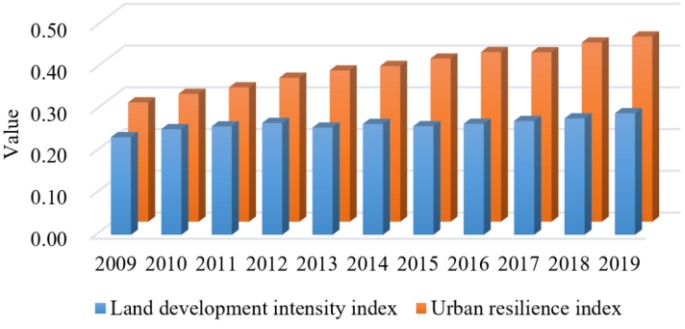

**Figure 3.** Land development intensity and urban resilience of the Yangtze River Delta urban agglomeration in 2009–2019.

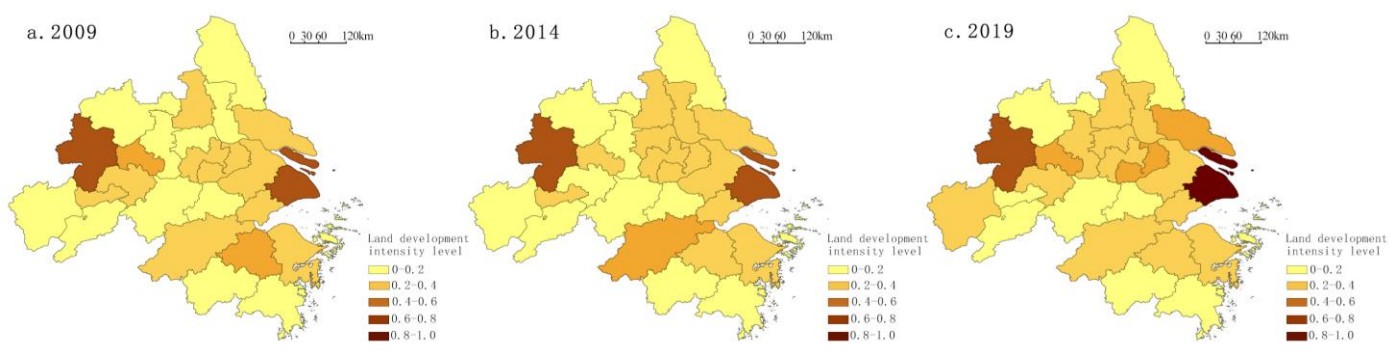

**Figure 4.** Spatial pattern of land development intensity in the Yangtze River Delta urban agglomeration in 2009–2019.

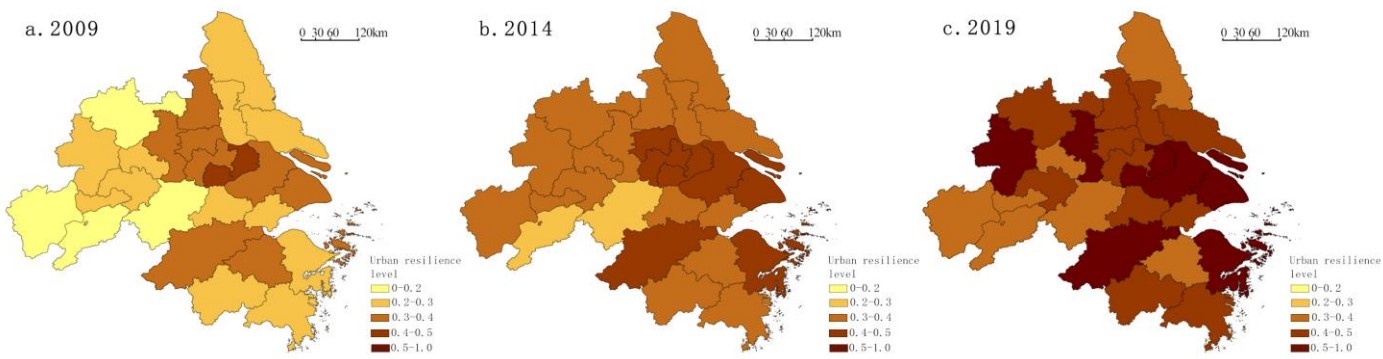

**Figure 5.** Spatial pattern of urban resilience in the Yangtze River Delta urban agglomeration in 2009–2019.

### 3.1.1. Overview of Land Development Intensity Level

From the temporal perspective, the land development intensity of the Yangtze River Delta urban agglomeration showed a fluctuating upward trend from 2009 to 2019: it first increased from 2009 to 2012, followed by a fluctuating downward trend from 2012 to 2016, and finally increased gradually after 2016. The average level of land development intensity increased by about 26% from 2009 (0.2338) to 2019 (0.2911). From the spatial perspective, the land development intensity of the Yangtze River Delta urban agglomeration exhibited obvious spatial differentiation, forming two cores with Shanghai and Hefei as the center, and gradually decreasing from the center to the periphery with an obvious circle-layered pattern. The areas with higher values include Shanghai and Hefei, as well as Hangzhou, Shaoxing, Wuxi, Nantong and Ma'anshan. These cities are more attractive in terms of population and capital due to their good economic foundation, superior location and transportation, which contribute to their high levels of land development intensity. The areas with lower values include Yancheng, Chuzhou, Anqing, Chizhou, Jinhua, Taizhou, Zhousha, Xuancheng and some other cities, which may be mainly attributed to their limited economic and industrial development and thus less attraction in terms of population and capital due to the restriction of natural geography and traffic location as well as the "siphoning effect" of surrounding core cities.

### 3.1.2. Overview of Urban Resilience Level

From the temporal perspective, the level of urban resilience of the Yangtze River Delta urban agglomeration increased significantly from 2009 to 2019. The average level of urban resilience continuously increased by about 52% from 2009 (0.2858) to 2019 (0.4436). As the Yangtze River Delta urban agglomeration has entered the post-urbanization period, there have been continuous increases in the investment in high-quality urbanization. The level of urban resilience has been greatly enhanced by the optimization

of urban ecological green environment, the improvement of public management and the construction of public service facilities and infrastructure. From the spatial perspective, the urban resilience of the Yangtze River Delta urban agglomeration shows significant variations. The areas with higher values are mainly provincial capital cities such as Shanghai, Nanjing, Hangzhou and Hefei and some regional core cities such as Suzhou, Wuxi and Ningbo, which have higher levels of urbanization and better economic foundation. In the process of urban development, these cities generally pay more attention to the improvement of urban quality and ecological environment. Urban public service facilities and infrastructure construction are usually ahead of schedule, which contributes to higher levels of urban resilience. Areas with lower values mainly include Yancheng, Xuancheng and some other cities, where the economic foundation is relatively poor, with imperfect supporting facilities of urban public services and infrastructure.

### 3.2. Spatio-Temporal Characteristics of Coupling Coordination Degree between Land Development Intensity and Urban Resilience

#### 3.2.1. Temporal Evolution of Coupling Coordination Degree

The coupling coordination degree model was used to determine the coupling coordination between land development intensity and the urban resilience of cities in the Yangtze River Delta from 2009 to 2019. From a temporal perspective, the average coupling coordination degree increased from 0.5177 in 2009 to 0.6626 in 2019, developing from bare coordination to moderate coordination in general. In addition, the coupling coordination degree increased in all cities, with the lowest value increasing from 0.1724 to 0.3724 and the highest value increasing from 0.7834 to 0.9814. There was a continuous decrease in the number of cities in the disordered stage and an increase in the number of cities with high-level coordination (Figure 6).

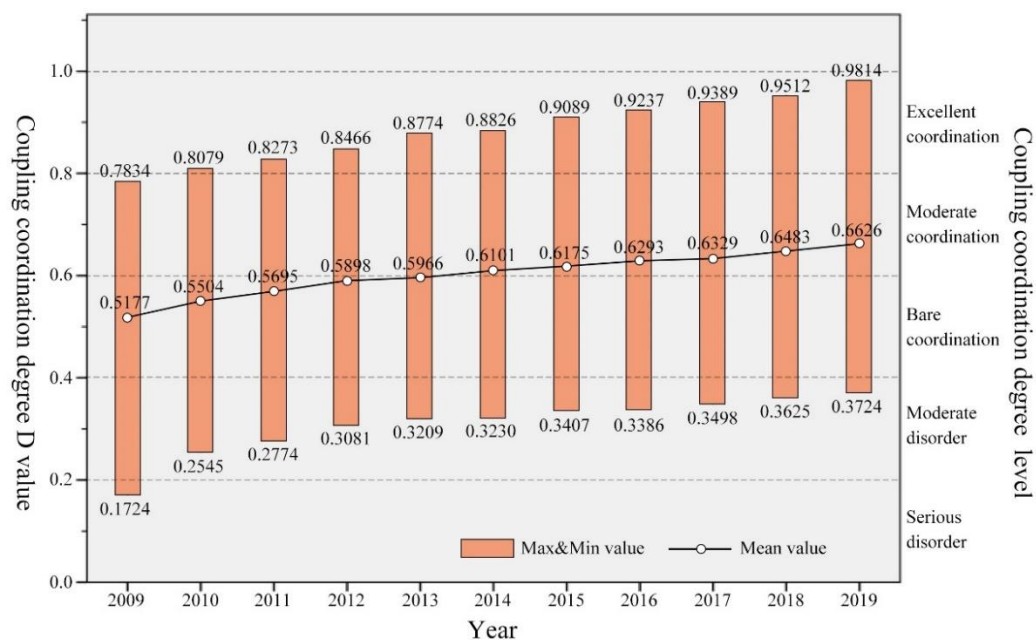

**Figure 6.** Temporal evolution of coupling coordination degree between land development intensity and urban resilience in the Yangtze River Delta urban agglomeration in 2009–2019.

Furthermore, kernel density estimation was carried out with the coupling coordination degree between land development intensity and urban resilience in 2009, 2014 and 2019 to better explore the temporal evolution trend of the coordinated development of the two systems and different cities (Figure 7). (1) In terms of the location of the kernel density curve, the curve in the study years shifted to the right in turn with the narrowing of the shift interval, indicating continuous increases in the coupling

coordination degree of land development intensity and urban resilience during the study period, but the increasing degree declined to some extent with time. (2) In terms of the shape of the kernel density curve, the peak height of the curve showed a gradual upward trend, and the rising trend gradually slowed down in the study years. The cure shape gradually changed from "wide peak" to "sharp peak" with significant narrowing of the distribution range. These results suggest that there are significant differences in the coupling coordination degree of land development intensity and urban resilience among different cities at the early stage of the study, but the differences are gradually decreased with time. (3) As for the tail of the curve, the left-side tail was shortened and the right-side tail was prolonged in the study years, demonstrating that the proportion of cities with low coupling coordination degrees was decreasing, while that of cities with high coupling coordination degrees was increasing with time.

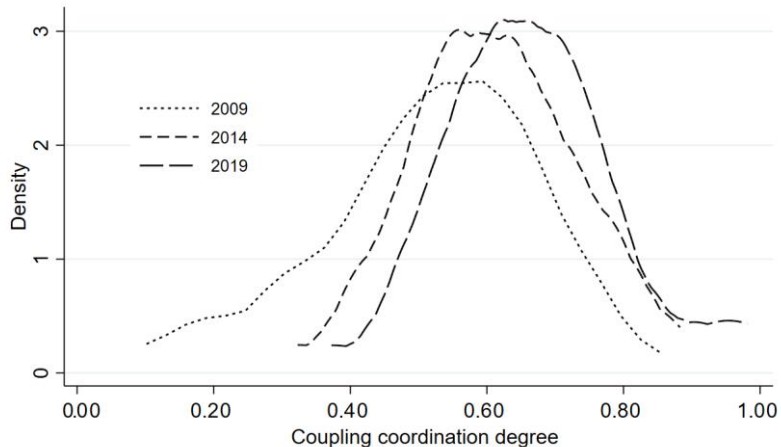

**Figure 7.** Kernel density estimation of coupling coordination degree between land development intensity and urban resilience in the Yangtze River Delta urban agglomeration in 2009–2019.

### 3.2.2. Spatial Differentiation Characteristics of Coupling Coordination Degree

To better explore the spatio-temporal differentiation characteristics of the coupling coordination degree between land development intensity and urban resilience in the Yangtze River Delta urban agglomeration, we selected 2009, 2014 and 2019 as time nodes and combined the classification standards described above to visualize the spatial distribution of the coupling coordination degree of these two systems in the selected years through the ArcGIS10.6 software (Figure 8).

As a whole, from 2009 to 2019, the coupling coordination degree of land development intensity and urban resilience of cities in the Yangtze River Delta urban agglomeration increased to different degrees. In terms of spatial distribution, coastal cities and cities along the Yangtze River had high coupling coordination degrees, while other cities were characterized by low coupling coordination degrees, forming a "T" shape distribution pattern.

Specifically, in 2009, the coupling and coordinated development level of land development intensity and the urban resilience of the urban agglomeration in the Yangtze River Delta was generally low, and most cities were in the stage of bare coordination. From the perspective of spatial distribution, three high-value regions (moderate coordination) were formed with Shanghai–Suzhou–Wuxi–Changzhou, Hangzhou–Shaoxing and Hefei as the core. In addition to the above cities, other cities in Jiangsu Province and Zhejiang Province were in the stage of bare coordination. Meanwhile, Anqing, Chizhou and Chuzhou in Anhui Province were in the stage of moderate disorder, and Xuancheng was still in the stage of serious disorder. In 2014, the gap between cities in the coupling coordination degree was gradually narrowed, but the majority of cities

were still in the stage of bare coordination. The coupling coordination degree between the two systems was improved significantly, and the regions with high values gradually gathered, forming a banded distribution pattern along the Yangtze River. Among them, Shanghai had developed to the stage of excellent coordination. Suzhou, Wuxi, Changzhou and Zhenjiang in Jiangsu Province, Hangzhou, Jiaxing and Ningbo in Zhejiang Province, and Hefei, Ma'anshan and Tongling in Anhui Province were in the moderate coordination stage. Most other cities were in the bare coordination stage except for Anqing, which was still in the moderate disorder stage. In 2019, the coupling coordination degree between land development intensity and urban resilience showed great improvement, and most cities were in the stage of moderate coordination. Coastal cities and cities along the Yangtze River almost developed to moderate coordination and the excellent coordination stage. Cities such as Yancheng, Huzhou, Jinhua, Chuzhou, Xuancheng and Tongling were in the stage of bare coordination, while Anqing in Anhui Province was still in the stage of moderate disorder. To some extent, these results suggest that provincial capital cities and economically developed cities generally have higher coupling coordination degrees between land development intensity and urban resilience, which can also drive the progress of the surrounding cities to gradually form an agglomeration area with high coupling coordination degrees.

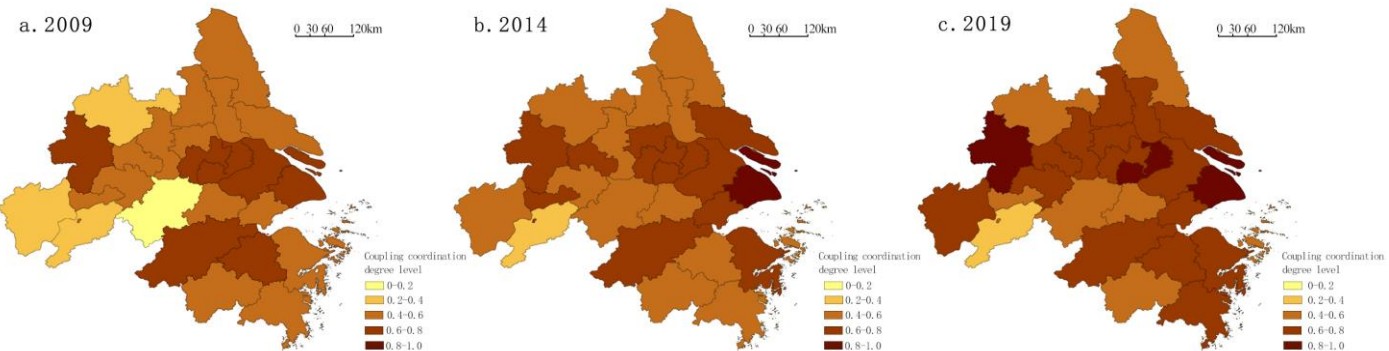

**Figure 8.** Spatial pattern of coupling coordination degree between land development intensity and urban resilience in the Yangtze River Delta urban agglomeration in 2009–2019. 0.0–0.2 represents serious disorder; 0.2–0.4 represents moderate disorder; 0.4–0.6 represents bare coordination; 0.6 - 0.8 represents moderate coordination; 0.8–1.0 represents excellent coordination.

### 3.2.3. Characteristics and Changes of Coupling Coordinated Development Types

Despite the constant improvement of the coupling coordination degree between land development intensity and urban resilience in the Yangtze River Delta urban agglomeration, there is still spatio-temporal heterogeneity in coupling coordinated development types among different cities (Table 2). In general, from 2009 to 2019, the majority of cities were characterized by "lagging behind in land development intensity". Among them, Nantong, Shaoxing, Tongling and other cities developed from the initial "lagging behind in urban resilience" to "lagging behind in land development intensity". On the one hand, these results indicate that after entering into the post-urbanization period, all cities paid more attention to the construction of urban resilience systems while pursuing the improvement of land development intensity, which greatly promotes their coordinated development. On the other hand, in the process of urban land development, infrastructure construction is usually the priority of investment, while population density and land economic benefits, which are generally used to measure the land development intensity, will lag behind for a certain period of time. Therefore, the level of land development intensity generally lags behind that of urban resilience. In addition, these results also reflect the fact that the land development and utilization of most cities in the Yangtze River Delta urban agglomeration are still relatively extensive. Therefore, it is

necessary to further improve the intensity and efficiency of land use to release the value of land.

In addition, Shanghai, Hefei and Ma'anshan always belonged to the "lagging behind in urban resilience" category during the study period. As a national central city and an important international financial and trade center, Shanghai has a high concentration of population and capital, and the land development intensity has always been maintained at a high level. Similarly, as the capital of Anhui Province, Hefei has a high population density. Particularly in recent years, by taking spillover industries from the core area of the Yangtze River Delta, its construction land has been expanded rapidly, resulting in high levels of land development intensity. Although the urban resilience level significantly improved in the above two cities during the study period, it still lags behind the high-intensity land development and utilization level. As a traditional heavy industrial city with steel as the leading industry along the Yangtze River, Ma'anshan has great pressure of energy and environmental pollution. Moreover, it has relatively low investment in scientific research and innovation and relatively weak construction of public service facilities and infrastructure. As a result, its level of urban resilience always lags behind the level of land development intensity.

**Table 2.** Coupling coordination degrees and types of land development intensity and urban resilience in the Yangtze River Delta urban agglomeration in 2009–2019.

| City | 2009 | | | | | 2019 | | | | |
|------|------|------|------|------|------|------|------|------|------|------|
| | Urban Resilience Level | Land Development Intensity | Coupling Coordination D Value | Coupling Coordination Level | Coupling Coordination Type | Urban Resilience Level | Land Development Intensity | Coupling Coordination D Value | Coupling Coordination Level | Coupling Coordination Type |
| Shanghai | 0.3661 | 0.6418 | 0.7834 | Moderate coordination | Urban resilience lags behind | 0.5504 | 0.9128 | 0.9814 | Excellent coordination | Urban resilience lags behind |
| Nanjing | 0.3062 | 0.1920 | 0.5407 | Bare coordinate | Land development intensity lags behind | 0.5130 | 0.2625 | 0.7020 | Moderate coordination | Land development intensity lags behind |
| Wuxi | 0.4635 | 0.2703 | 0.6842 | Moderate coordination | Land development intensity lags behind | 0.5261 | 0.4372 | 0.8045 | Excellent coordination | Land development intensity lags behind |
| Changzhou | 0.3851 | 0.2682 | 0.6408 | Moderate coordination | Land development intensity lags behind | 0.4269 | 0.3362 | 0.7030 | Moderate coordination | Land development intensity lags behind |
| Suzhou | 0.3756 | 0.3162 | 0.6620 | Moderate coordination | Land development intensity lags behind | 0.5751 | 0.2957 | 0.7500 | Moderate coordination | Land development intensity lags behind |
| Nantong | 0.2524 | 0.3333 | 0.5723 | Bare coordinate | Urban resilience lags behind | 0.4441 | 0.4373 | 0.7611 | Moderate coordination | Land development intensity lags behind |
| Yancheng | 0.2122 | 0.1427 | 0.4244 | Bare coordinate | Land development intensity lags behind | 0.3718 | 0.1359 | 0.5332 | Bare coordinate | Land development intensity lags behind |
| Yangzhou | 0.3500 | 0.2316 | 0.5967 | Bare coordinate | Land development intensity lags behind | 0.4120 | 0.2752 | 0.6604 | Moderate coordination | Land development intensity lags behind |
| Zhenjiang | 0.3141 | 0.1544 | 0.5169 | Bare coordinate | Land development intensity lags behind | 0.4556 | 0.2471 | 0.6651 | Moderate coordination | Land development intensity lags behind |
| Taizhou | 0.2995 | 0.1923 | 0.5361 | Bare coordinate | Land development intensity lags behind | 0.4044 | 0.2334 | 0.6296 | Moderate coordination | Land development intensity lags behind |
| Hangzhou | 0.3564 | 0.2994 | 0.6407 | Moderate coordination | Land development intensity lags behind | 0.5545 | 0.3578 | 0.7778 | Moderate coordination | Land development intensity lags behind |
| Ningbo | 0.2997 | 0.2270 | 0.5592 | Bare coordinate | Land development intensity lags behind | 0.5088 | 0.2960 | 0.7217 | Moderate coordination | Land development intensity lags behind |
| Jiaxing | 0.2693 | 0.2479 | 0.5467 | Bare coordinate | Land development intensity lags behind | 0.4511 | 0.3360 | 0.7161 | Moderate coordination | Land development intensity lags behind |
| Huzhou | 0.2760 | 0.0743 | 0.4073 | Bare coordinate | Land development intensity lags behind | 0.4346 | 0.1486 | 0.5759 | Bare coordinate | Land development intensity lags behind |

| | | | | | | | | | | |
|---|---|---|---|---|---|---|---|---|---|---|
| Shaoxing | 0.3099 | 0.4044 | 0.6552 | Moderate coordination | Urban resilience lags behind | 0.3879 | 0.2673 | 0.6420 | Moderate coordination | Land development intensity lags behind |
| Jinhua | 0.2903 | 0.0700 | 0.4097 | Bare coordinate | Land development intensity lags behind | 0.4307 | 0.0950 | 0.5126 | Bare coordinate | Land development intensity lags behind |
| Zhoushan | 0.3019 | 0.0636 | 0.4062 | Bare coordinate | Land development intensity lags behind | 0.5047 | 0.1067 | 0.5563 | Bare coordinate | Land development intensity lags behind |
| Taizhou | 0.2897 | 0.1215 | 0.4710 | Bare coordinate | Land development intensity lags behind | 0.4785 | 0.1719 | 0.6169 | Moderate coordination | Land development intensity lags behind |
| Hefei | 0.2942 | 0.6064 | 0.7103 | Moderate coordination | Urban resilience lags behind | 0.5274 | 0.7631 | 0.9257 | Excellent coordination | Urban resilience lags behind |
| Wuhu | 0.2918 | 0.2412 | 0.5616 | Bare coordinate | Land development intensity lags behind | 0.4130 | 0.2798 | 0.6638 | Moderate coordination | Land development intensity lags behind |
| Ma'anshan | 0.2334 | 0.4549 | 0.5964 | Bare coordinate | Urban resilience lags behind | 0.3672 | 0.4247 | 0.7071 | Moderate coordination | Urban resilience lags behind |
| Tongling | 0.2327 | 0.2428 | 0.5086 | Bare coordinate | Urban resilience lags behind | 0.3529 | 0.1714 | 0.5547 | Bare coordinate | Land development intensity lags behind |
| Anqing | 0.1732 | 0.1440 | 0.3749 | Moderate disorder | Land development intensity lags behind | 0.3404 | 0.2487 | 0.6012 | Moderate coordination | Land development intensity lags behind |
| Chuzhou | 0.1769 | 0.0437 | 0.2808 | Moderate disorder | Land development intensity lags behind | 0.4143 | 0.1512 | 0.5691 | Bare coordinate | Land development intensity lags behind |
| Chizhou | 0.1916 | 0.0108 | 0.2022 | Moderate disorder | Land development intensity lags behind | 0.3369 | 0.0385 | 0.3724 | Moderate disorder | Land development intensity lags behind |
| Xuancheng | 0.1184 | 0.0828 | 0.1724 | Serious disorder | Land development intensity lags behind | 0.3503 | 0.1395 | 0.5251 | Bare coordinate | Land development intensity lags behind |

### 3.3. Influencing Factors on Coupling Coordination Degree between Land Development Intensity and Urban Resilience

#### 3.3.1. Variable Selection

There are various factors that affect the coordinated development of land development intensity and urban resilience, which are associated with the level of economic and industrial development as well as scientific and technological innovation, social management and infrastructure. Based on the existing literature [28,37,38], and combined with the actual situation of the development of the Yangtze River Delta urban agglomeration, we explored the influence of land economic benefit, industrial green development, scientific and technological innovation, social management and infrastructure. Specifically, GDP per $km^2$ land in the municipal district ($x1$) was selected to represent the land economic benefit level; the industrial $SO_2$ emission per 10,000 yuan GDP ($x2$) was used to represent the industrial green development level; the proportion of science and technology expenditure in financial expenditure ($x3$) was selected to represent the scientific and technological innovation level; the number of public management and social organization personnel per 10,000 people ($x4$) was used to represent the social management level; the density of water supply pipelines in a built-up area ($x5$) was selected to represent the infrastructure construction level.

#### 3.3.2. Results Analysis

The coupling coordination degree (*D*) value between land development intensity and urban resilience was taken as the explained variable, and the selected influencing factors were taken as the explanatory variables. In order to eliminate the influence of heteroscedasticity, the natural logarithm was taken for all non-proportional variables. The Stata 15.0 software was used for model estimation, and the panel Tobit regression results are reported in Table 3.

The log likelihood value (681.56) indicates that the model has good fitting with a high Wald test value (2087.57), rejecting the hypothesis that there is no individual effect. Therefore, the Tobit regression model of panel random effect can effectively reflect the

influence of various factors on the coupling coordination degree between land development intensity and urban resilience.

From the regression results, all variables pass the significance level test of 1%. Among them, GDP per $km^2$ land in the municipal district, proportion of science and technology expenditure in financial expenditure, the number of public management and social organization personnel per 10,000 people and the density of water supply pipelines in a built-up area have positive impacts on the coupling coordination degree between land development intensity and urban resilience, while the industrial $SO_2$ emission per 10,000 yuan GDP has a negative impact. (1) In terms of land economic benefits, GDP per $km^2$ land in the municipal district is an important factor to measure the level of land development intensity. In addition, a higher level of land economic benefit also reflects a higher level of urban economic development, and correspondingly, a stronger ability to deal with risks. Therefore, the improvement of land economic efficiency can significantly promote the coordinated development of land development intensity and urban resilience. (2) In terms of industrial green development, industrial $SO_2$ emission per 10,000 yuan GDP negatively affects the coupling coordination degree between land development intensity and urban resilience, indicating that the industrial green development level in the Yangtze River Delta urban agglomeration is still relatively low, and the environmental pressure caused by high consumption, high pollution and high emission levels in the process of industrial development hinders the coordinated development of land development intensity and urban resilience to some extent. (3) In terms of scientific and technological innovation, an increase in scientific and technological investment can significantly promote the coordinated development of land development intensity and urban resilience. On the one hand, science and technology, as the primary productive force, can promote the intensive and efficient utilization of land resources. On the other hand, investment in science and technology can also enhance the comprehensive ability of urban management, medical treatment, innovation and other aspects, so as to improve the ability of a city to deal with risks. (4) In terms of social management, an increase in the number of personnel in public management and social organizations can significantly promote the coordinated development of land development intensity and urban resilience, mainly because a higher level of public management can support the high-intensity and high-efficiency operation of a city. Even when faced with great risks, it can maintain the normal operation of social order to the greatest extent. (5) In terms of infrastructure, the density of water supply pipelines in a built-up area has a significant positive impact on the coupling coordination between land development intensity and urban resilience, mainly because increasing infrastructure construction can more effectively support the land development intensity. In addition, it can also improve the urban engineering resilience, so as to promote the coordinated development of land development intensity and urban resilience.

**Table 3.** Tobit regression results of influencing factors on the coupling coordination degree between land development intensity and urban resilience in the Yangtze River Delta urban agglomeration.

| Variable | Regression Coefficient | Standard Deviation | *Z* Score | *P* Value |
|---|---|---|---|---|
| GDP per $km^2$ land in municipal district (*x*1) | 0.1225 | 0.0048 | 25.75 | 0.000 |
| Industrial $SO_2$ emission per 10,000 yuan GDP (*x*2) | −0.0079 | 0.0013 | −6.06 | 0.000 |
| Proportion of science and technology expenditure in financial expenditure (*x*3) | 0.0305 | 0.0045 | 6.76 | 0.000 |
| Number of public management and social organization personnel per 10,000 people (*x*4) | 0.0232 | 0.0081 | 2.85 | 0.004 |
| Density of water supply pipeline in built-up area (*x*5) | 0.0430 | 0.0051 | 8.38 | 0.000 |
| Constant term (*_cons*) | −0.7896 | 0.0494 | −15.99 | 0.000 |

## 4. Conclusions and Suggestions

*4.1. Conclusions*

Modern society is entering a "risk-oriented society". Hence, it is very important to enhance the coordination between land development intensity and urban resilience and build sustainable and safe cities. This paper constructed a comprehensive evaluation index system to measure the level of land development intensity and urban resilience and analyzed their characteristics in the Yangtze River Delta urban agglomeration. Furthermore, the temporal and spatial evolution characteristics of the coupling coordination degree between these two systems were analyzed, and then, the influencing factors were determined through the panel Tobit regression model. The following conclusions can be drawn:

(1) From 2009 to 2019, the levels of land development intensity and urban resilience in the Yangtze River Delta urban agglomeration showed a general upward trend. The average level of land development intensity increased from 0.2338 to 0.2911, forming a circle-layered pattern with Shanghai and Hefei as the center. The average level of urban resilience continuously rose from 0.2858 to 0.4436. There was significant spatial differentiation, with the formation of a high-value region by Shanghai, Nanjing, Hangzhou, Hefei and other provincial capitals and regional central cities such as Suzhou, Wuxi and Ningbo. On the whole, the level of land development intensity lagged behind that of urban resilience, indicating that the land development and utilization of the Yangtze River Delta urban agglomeration is still relatively extensive. On the premise of satisfying urban resilience, it is necessary to further improve the intensity and efficiency of land use.

(2) From 2009 to 2019, the average coupling coordination degree between land development intensity and urban resilience of the Yangtze River Delta urban agglomeration increased from 0.5177 to 0.6626. The level of coordination between the two systems was continuously improved and generally developed from bare coordination to moderate coordination. In addition, the difference among cities in the coupling coordination degree between the two systems was gradually decreased. The regions with high coupling coordination degrees formed a "T"-shaped structure along the coast and the Yangtze River, while the peripheral cities had relatively low coordination degrees, forming an obvious circle-layered pattern. In terms of the type of coupled coordinated development, the "lagging behind in land development intensity" category accounts for the majority, but Shanghai, Hefei and Ma'anshan have always been categorized as "lagging behind in urban resilience".

(3) The panel Tobit regression model analysis revealed that land economic benefit, industrial green development, scientific and technological innovation, social management and infrastructure have significant impacts on the coupling and coordinated development of land development intensity and urban resilience in the Yangtze River Delta urban agglomeration. Among various factors, GDP per $km^2$ land in the municipal district, the proportion of science and technology expenditure in financial expenditure, number of public management and social organization personnel per 10,000 people and the density of water supply pipelines in a built-up area have positive impacts on the coupling coordination degree of land development intensity and urban resilience, while the industrial $SO_2$ emission per 10,000 yuan GDP has a negative impact.

*4.2. Policy Suggestions*

As a world-class urban agglomeration, the Yangtze River Delta urban agglomeration is characterized by the high convergence of population, capital and land. The coordinated development of land development intensity and urban resilience is directly related to the safe and sustainable development of urban agglomerations in the future. Based on the research findings for the spatio-temporal differentiation characteristics and influencing factors of the coupling coordination degree between land development intensity and urban resilience in the Yangtze River Delta urban agglomeration from 2009 to 2019, this

paper puts forward the following policy suggestions: (1) The radiation and spillover effects of central cities such as Shanghai, Hangzhou, Nanjing and Hefei should be fully acknowledged to promote the coordinated development of regional land development intensity and urban resilience. (2) Different policies are required by different cities. Differential measures should be adopted by cities with different types and characteristics to improve urban land development intensity and urban resilience according to their own developmental stages. For cities categorized as "lagging behind in urban resilience", it is necessary to increase investment and construction in ecological environment, social management, scientific and technological innovation and infrastructure, so as to promote urban resilience. For cities categorized as "lagging behind in land development intensity", it is necessary to control the expansion of construction land, improve the development intensity of stock land and promote more intensive, efficient and sustainable land development. (3) In the process of urbanization, the government should strengthen the intensive use of land, improve the land economic benefits, adjust the industrial structure and promote the green development of industry. In addition, policies should be made to increase investment in scientific and technological innovation, improve the level of social public management and infrastructure and promote the coordinated development of land development intensity and urban resilience.

Due to the complexity of the interaction between land development intensity and urban resilience and the limitation of data and materials, this paper only included a preliminary discussion on the interaction and influencing factors of the land development intensity and urban resilience of 26 cities in the Yangtze River Delta urban agglomeration from the perspective of coupling coordination. It is surely necessary to further optimize the index system in the future and include more perspectives and compare different urban agglomerations, so as to more systematically explore the internal mechanisms and influencing factors of land development intensity and urban resilience.

**Author Contributions:** Conceptualization, C.C. and X.L.; methodology, C.C.; software, C.C.; validation, Y.P.; formal analysis, Y.P.; investigation, C.C.; resources, C.C.; data curation, Y.P.; writing—original draft preparation, C.C.; writing—review and editing, C.C. and T.Y.; visualization, C.C.; supervision, X.L.; project administration, X.L.; funding acquisition, X.L. All authors have read and agreed to the published version of the manuscript.

**Funding:** This research was funded by the National Natural Science Foundation of China (71974071; 42171286).

**Institutional Review Board Statement:** Not applicable.

**Informed Consent Statement:** Not applicable.

**Data Availability Statement:** The data presented in this study are available on request from the corresponding author.

**Conflicts of Interest:** The authors declare no conflict of interest.

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
