# Peer review of "Coupling Coordination and Influencing Factors of Land Development Intensity and Urban Resilience of the Yangtze River Delta Urban Agglomeration"

_water, doi:10.3390/w14071083_

Round 1

Reviewer 1 Report

Dear authors,

the idea of analyzing urban resilience as a function of different factors is an interesting topic. The paper is well written and well structured. Objective of the study is clear although exposure can be improved.

The title is too long, we need to think of a more concise title that captures the reader's attention

In the introduction it is necessary to insert how the article is structured and highlight the study objective. If I read the whole paper, the objective is understood, but in the introduction it is necessary to insert better detail.

Can the proposed approach also be applied to other case studies?

Reviewer 2 Report

The article deals with a topic of the urban resilience, which is important especially in the context of adoption to climate change. Perhaps it would be worth developing this thread in the justification for undertaking the research topic.

Comments:

  1. The aim of the paper should be clearly defined.
  2.  Referring to Figure 1, dividing urban resilience into four subsystems is not explained in the text. According to pillars of sustainable development there er three main subsystems: economy, society and environment. Introducing the fourth subsystem should be explained.
  3. Citing Authors understanding of ecological resilience: ‘…is the basis of urban sustainable development, reflecting the service function of an urban ecosystem and the level of green development’. Taking into account this explanation, why among seven indicators covering this subsystem, two of them are connected directly to risk and endangering (wastewater emission and SO2 emission)? They do not build the urban ecological resilience.

Author Response

This manuscript is a resubmission of an earlier submission. The following is a list of the peer review reports and author responses from that submission.